# Fabrication of MgO-Y_2_O_3_ Composite Nanopowders by Combining Hydrothermal and Seeding Methods

**DOI:** 10.3390/ma16010126

**Published:** 2022-12-23

**Authors:** Shangyu Yang, Hao Lan, Xiaoming Sun, Shaowei Feng, Weigang Zhang

**Affiliations:** 1School of Rare Earths, University of Science and Technology of China, Hefei 230026, China; 2 Ganjiang Innovation Academy, Chinese Academy of Sciences, Ganzhou 341119, China; 3Institute of Process Engineering, Chinese Academy of Sciences, Beijing 100190, China; 4School of Chemical Engineering, University of Chinese Academy of Sciences, Beijing 100049, China

**Keywords:** MgO-Y_2_O_3_ nanopowders, hydrothermal, seeding, hot-pressing, transparent ceramic

## Abstract

In this study, the combination of hydrothermal technique and seed-doping method was conducted to coordinately control the formation of fine MgO-Y_2_O_3_ powders, which are promising mid-infrared materials applied to hypersonic aircraft windows due to their excellent infrared transmissions over wide regions. Y(NO_3_)_3_·6H_2_O, Mg(NO_3_)_2_·6H_2_O, Y_2_O_3_ seeds and MgO seeds were used as raw materials to prepare the MgO-Y_2_O_3_ composite powders (50:50 vol.%), and the influences of the seed contents and hydrothermal treatment temperatures on the final powders and hot-pressed ceramics were investigated by XRD, SEM and TEM techniques. The results show that powders with a seed content of 5% that are hydrothermally synthesized at 190 °C can present a better uniformity and dispersion with a particle size of ~125 nm. Furthermore, the ceramics prepared with the above powders displayed a homogenous two-phase microstructure, fewer pores and a fine grain size with Y_2_O_3_ of ~1 µm and MgO of ~620 nm. The present study may open an avenue for developing transparent ceramics based on MgO-Y_2_O_3_ nanopowders prepared by hydrothermal technique.

## 1. Introduction

Transparent ceramics are a candidate for many applications, such as mid-infrared window materials, owing to their excellent mechanical, thermal and optical properties [1,2,3,4,5,6]. The mid-infrared window materials that are applied to hypersonic aircraft windows require excellent service performances at a high temperature [7,8,9]. In recent years, many investigations have focused on Y_2_O_3_-based transparent ceramics, which are proven to be promising window materials because of their wide band transmission, low infrared emissivity, low scattering [10,11,12] and good mechanical properties [13,14,15,16]. Among these yttrium-oxide-based transparent ceramics, MgO-Y_2_O_3_ has been reported to present a better performance than pure Y_2_O_3_ ceramics or pure MgO ceramics, owing to its better optical transmittance, thermal shock resistance and mechanical strength [17,18,19]. This is because the addition of MgO will be helpful in inhibiting grain growth, and then the grain size of the transparent ceramic will be smaller than that of the yttrium ceramics without MgO addition [20,21,22,23]. Moreover, the fine grain size of MgO-Y_2_O_3_ powders, which are essential for obtaining fine grains and uniform microstructures, will positively influence the final sinterability. During the past years, many efforts have been devoted to the study and production of high-quality MgO-Y_2_O_3_ powders.

Methods to produce fine MgO-Y_2_O_3_ composite powders, such as citrate-nitrate combustion [24], chemical co-precipitation [25], sol-gel [26], bioorganic [27] and hydrothermal methods [28], have been proposed. In general, the above methods, such as the chemical co-precipitation method and the citrate-nitrate combustion method, usually take a long reaction time and introduce impurities in the formation of MgO-Y_2_O_3_ nanocrystals [29]. In addition, the powders produced by sol-gel, combustion and bioorganic methods are likely to agglomerate [21,22,23,24,25,26,27,28,29,30]. The hydrothermal method is proven to be an effective method for the synthesis of fine composite powders due to the extensive milling, high phase purity and narrow particle size distribution [31]. As reported before, the hydrothermal technique can not only provide a homogeneous nucleation process to control the size and morphology of crystallites but also significantly reduce the aggregation levels. Moreover, it has been demonstrated that the seeding method represents a unique way to produce less agglomerate nanopowders to obtain fine Y_2_O_3_ powders [32].

Thus, in this study, the combination of the hydrothermal technique and seed-doping method to coordinately control the formation of fine MgO-Y_2_O_3_ powders was implemented. The influences of seed concentrations and hydrothermal temperatures on the morphologies of MgO-Y_2_O_3_ powders were systemically studied. The composite powders were then sintered by using a hot-pressing (HP) sintering technique [33,34,35].

## 2. Materials and Methods

### 2.1. Powders

Nanopowders with magnesium oxide and yttrium oxide at a 50:50 vol ratio were prepared via hydrothermal method. Yttrium nitrate hexahydrate [Y(NO_3_)_3_·6H_2_O, purity: 99.99%, Aladin, Shanghai, China], magnesium nitrate hexahydrate [Mg(NO_3_)_2_·6H_2_O, purity:99%, Shanghai test chemical reagent, China] and ammonia were used as the starting materials. Then, Y(NO_3_)_3_·6H_2_O and Mg(NO_3_)_2_·6H_2_O with a Y_2_O_3_-to-MgO volume ratio (Y_2_O_3_:MgO = 50:50) were dissolved in distilled deionized water. After that, the solution was clear and colorless. Then, the nitrate mixtures were stirred for 10 min, and the molar ratio of the ammonia to the distilled deionized (D.I.) water was 0.1. The nitrate mixtures were added into the ammonia solution at room temperature. In this solution, the mixture was maintained at pH 12. Afterwards, the solution, washed twice with D.I. water, was transferred to a 500 mL beaker. Then, different ratios of MgO:Y_2_O_3_ seeds (v:v = 50:50) were added into the beaker. Y_2_O_3_ (purity: 99.99%, Hawk, Beijing, China) and MgO (purity: 99.99%, Hawk, China) were dispersed by ultrasound before adding them into the beaker. The beaker was put in a hydrothermal titanium kettle at 160, 190, 220, 230 °C for 120 min with different seed contents; hereafter, the samples with a 5% seed content are abbreviated as 160C-5S, 190C-5S, 220C-5S and 230C-5S, respectively. The heating rate of the hydrothermal kettle was 1.5 °C/min. We investigated the influence of the amounts of seeds and hydrothermal temperatures of the hydrothermal method on the properties of MgO-Y_2_O_3_ nanocomposites. The sample was cooled to room temperature, and then the hydrothermal products were evacuated and dried at 60 °C. They were then converted to white powders by placing them in a furnace, calcined at 1000 °C for 120 min with a heating rate of 10 °C/min and then cooled naturally to room temperature. Finally, MgO-Y_2_O_3_ powders were obtained.

### 2.2. HP Process

A hot-pressing sintering machine (Shanghai Chenrong Electric Furnace Co., Ltd., Shanghai, China) was utilized in this study. The nanopowders were poured into a graphitic die, and the graphitic foil was used to isolate the nanopowders. The synthesized nanocomposite was consolidated in the HP system under vacuum. The specimen was heated from room temperature to 1000 °C with a heating rate of 10 °C/min and then heated to 1550 °C with a heating rate of 5 °C/min and kept 120 min before being cooled to room temperature. Meanwhile, the specimens were pressed at a rate of 0.1 MPa/s from 0 MPa to 40 MPa starting at 1200 °C by gradient pressurization. Before the pressure reached 40 MPa, the pressure was maintained for 18 min during every 10 MPa. As a result, the specimens were held at 1550 °C for 120 min with a pressure of 40 MPa. During the cooling process, the pressure gradually decreased from 40 MPa to 0 MPa until 1200 °C. After removing the graphite foil from the surface of the compressed MgO-Y_2_O_3_ disk, the sample was annealed at 1250 °C for 360 min in air, and the disk was carefully polished before using SEM to observe the details of the sample surface.

### 2.3. Characterization

The microstructures of the specimens were characterized by scanning electron microscopy (SEM, JSM-7001F+INCA X-MAX). The thermal characteristics of the hydrothermal powders were analyzed by thermogravimetric-differential scanning calorimetry (TG-DSC). X-ray diffraction (XRD, X’Pert PRO MPD; PANalytical B.V., Almelo, The Netherlands) was utilized to identify the phase formation of the powders and the sintered body using CuKα radiation (λ = 1.5406 Å, Generator voltage = 40 kV, Tube current = 40 mA). The XRD spectra were acquired by scanning over the angular range 2 θ = 5–90° at a continuous scan time and time per step = 15.24. Scherrer formula was used to estimate the average crystalline size. X-ray fluorescence (XRF, AXIOS, PANalytical B.V.) was used to roughly find the elemental composition of the powder. The average particle size was calculated based on the Brunauer–Emmett–Teller (BET, BET specific surface full-automatic physical adsorption analyzer, Quantachrome Instruments, Autosorb-iQ-3, Boynton Beach, FL, USA). The morphologies of the powders and MgO-Y_2_O_3_ ceramics were further observed by transmission electron microscopy (TEM, JEOL JEM-F200, Tokyo, Japan). Furthermore, TEM images with the selected area diffraction patterns of MgO-Y_2_O_3_ ceramics were observed.

## 3. Results and Discussion

### 3.1. Characterization of the MgO-Y_2_O_3_ Nanopowders

As can be seen in Figure 1, which shows a SEM image of the hydrothermal precursors, the powders were all characterized by a square flakiness, and the average particle size was normally below 200 nm. No other impurities could be found in the precursors.

The thermal characteristics of the hydrothermal precursors during the hydrothermal nanopowders’ synthesis were investigated by using the thermal analysis technique, and the TG-DSC curves of hydrothermal precursors are shown in Figure 2. The first weight-loss step over the temperature range ranging from room temperature to 200 °C, with a weight loss of about 6.15%, was due to absorbed water molecules [36,37,38]. The second weight loss was about 17.87%, from 200 °C to 490 °C, which could be the removal of hydroxyl. The last weight loss was about 7.77% from 490 °C to 1000 °C, which was caused by Y(NO_3_)_3_ decomposition and Mg(NO_3_)_2_ decomposition. Above 1000 °C, no significant weight loss was observed. The endothermic peak around 340 °C was ascribed to the decomposition of a small amount of nitrate in the mixture of water vapor and nitrate, and it may also be ascribed to the crystallization of the residual amorphous phase and formation of metal oxides [36,37,38].

The XRD patterns of the hydrothermal samples with different seed contents at 230 °C were analyzed, and the results are presented in Figure 3a. The diffraction peaks confirm the formation of MgO and Y_2_O_3_ structures for all samples and that they could all be readily indexed to the Y_2_O_3_ (JCPDS No. 01-079-1717) cubic phase and MgO (JCPDS No. 01-089-7746) cubic phase, indicating a complete synthesis of Y_2_O_3_-MgO particles under current experimental conditions. Other material structures or impurity phases are not observed. There are no obvious differences in the peak positions for all the synthesized powders. However, our careful analysis showed that a difference in the full width at the half maximum (FWHM) of the X-ray peaks of the synthesized particles can be detected. The FWHM values of 230C-5S (Y_2_O_3_ at 29° and MgO at 42°) are broader than the values of the other samples (see Figure 3b), indicating that the particles synthesized with 5% seed added have a finer crystallite size than others. In addition, the XRF analysis showed that the mass fractions of Y_2_O_3_ and MgO were 61.94% and 38.06%, respectively.

Figure 4 shows the SEM images of MgO-Y_2_O_3_ nanoparticles by using different seed contents hydrothermally synthesized at 230 °C. One can see that the obtained MgO-Y_2_O_3_ powders were separated with uniform particle sizes between 100 nm and 150 nm. The powders had a good crystallinity, without the detection of any other phases. One can note from Figure 4c that when the seed content reached 5%, the powders were better dispersed and less aggregated compared with seed contents of 0% (Figure 4a), 1% (Figure 4b) and 10% (Figure 4d). Furthermore, one can see that the nanopowders are flaky with an irregular morphology. In contrast to the 230C-0S (Figure 4a), the morphologies of the powders with different seed contents are composed of flake and some elliptical small particles.

The relationship between the mass percentage of seeds and the specific surface area (based on the BET analysis) of the nanocomposite powders hydrothermally synthesized at 230 °C is shown in Figure 5. The value of the specific surface area was 40.3 m^2^/g without adding seed. The composite nanopowders showed a specific surface area of 28.6 m^2^/g with the seed percentage of 1%. When the seed percentage reached 5%, the specific surface area was approximately 47.9 m^2^/g. However, the specific surface area decreased to 34.8 m^2^/g when the seed percentage increased to 10%. The above results came from the fact that a certain mass percentage of seeds may lead to an increase in the transformation kinetics of hydroxide to oxide during a hydrothermal reaction [32]. According to the SEM results shown in Figure 4, there was little difference in the average particle size of the composite powders; thus, the local agglomerations of the particles may be the main factor of the specific surface area. Overall, the specific surface area of the powder after hydrothermal treatment at 230 °C with a 5% seed content was the largest compared with other powders of different seed contents. Thus, considering the above results, in the following part we choose seed contents of 5% when changing the hydrothermal temperatures to further optimize the process.

Figure 6 shows the SEM images of the powders of 160C-5S, 190C-5S and 220C-5S. These composite powders consisted of local agglomerates of different shapes and sizes. The powders shown in Figure 6a,b formed short, solid cylinders, whereas the powders shown in Figure 6c formed a mixture of cylinders and flakes. When reaching 230 °C, the powders all grew into flakes. One can see that the short, solid cylinders were locally clustered together at 160 °C, while the aggregation of composite powders decreased at 190 °C. This indicates that a suitable hydrothermal temperature caused the composite powders’ good dispersion. As the temperature increased to 220 °C and 230 °C, the seed effect under these temperature conditions further caused the increase of nucleation. In addition, the increase of temperature may enlarge the precursor size, and the following calcination step will not reverse such a situation. Hence, a 5% seed addition and a hydrothermal synthesis at 190 °C of the powders are prone to be the most suitable parameters for less aggregation of composite powders in this study.

Since the powders’ particle size and morphology cannot be comprehensively detected in the SEM observations because of the local agglomeration of nanopowders, the morphology and crystallinity of the synthesized Y_2_O_3_-MgO nanoparticles were further investigated by TEM and SAED patterns. Figure 7 shows more detailed microstructures of the TEM micrograph for 190C-5S nanopowders. As can be seen in Figure 7a, each sample consisting of Y_2_O_3_ and MgO phases is homogeneously distributed, with a short rod-shaped morphology within a particle size of 110 nm. Based on the SAED analysis in Figure 7b–e, the indexed diffraction patterns reveal a cubic crystal structure. EDS spectra show that the Y_2_O_3_ and MgO particles were uniformly dispersed in the mixtures. EDS analysis also exhibits that the matrix mainly contains magnesium, yttrium and oxygen, without any other elements. Moreover, the elements are evenly dispersed in particles. MgO-Y_2_O_3_ particles found from SEM micrographs were further confirmed to be uniformly dispersed from the EDS analysis. Therefore, the present hydrothermal method can obtain homogeneous MgO-Y_2_O_3_ composite powders.

The Y(NO_3_)_3_·6H_2_O and Mg(NO_3_)_2_·6H_2_O with a Y_2_O_3_-to-MgO volume ratio (Y_2_O_3_:MgO = 50:50) were dissolved in distilled deionized water, resulting in Brucite which was generated when the distilled water reacted with MgO (periclase) and produced Mg(OH)_2_. Brucite can potentially result in volume expansions or “unsoundness”. In the present work, “soundness” can refer to the purity, crystallinity, grain size and volume-content stability of the production. The influence of MgO seeds in distilled water should be considered. Therefore, we plan to measure the volume stability of MgO-Y_2_O_3_ composites by using a Le Chatelier test in the future [39,40].

In the present study, we combined the hydrothermal technique and seed-doping method to coordinately control the formation of fine MgO-Y_2_O_3_ powders. On the one hand, the hydrothermal method can enhance the solubility and reactivity of MgO-Y_2_O_3_ precursors under high pressure and elevated temperatures in autoclaves. During this solvothermal process, anisotropically shaped MgO-Y_2_O_3_ particles may appear when the competitive balance between growth kinetics and crystallographic anisotropies is broken [30]. On the other hand, applying the MgO and Y_2_O_3_ double-seeding method led to an increase in transformation kinetics and to a decrease of the temperature at which a transition from hydroxide to oxide occurs during the hydrothermal reaction. Under this co-regulating condition, deagglomerated particles of the yttria and magnesium oxide can be produced at a relatively low temperature (190 °C) in this work.

### 3.2. Characterization of the MgO-Y_2_O_3_ Ceramics

In order to observe the microstructure of MgO-Y_2_O_3_ composite ceramics after hot-press sintering, including the grain size and distribution, the samples were thermally etched, and the microstructures and phase distributions of 230C-0S, 160C-5S, 190C-5S and 220C-5S are shown in Figure 8a–d, respectively. As can be seen, there is a clear quality difference between Y_2_O_3_ and MgO in all figures, where the darker one is MgO and the brighter one is Y_2_O_3_. Ma et al. [41] concluded that fine particles and uniform phase separation will produce fine particles after hot pressing, making the optimization of particle synthesis pretty important to improving mechanical and optical properties [41]. The introduction of MgO as the second phase will increase the strain and limit the microcrystalline growth [42]. As can be seen in Figure 8a of 230C-0S, without the addition of seeds, the phase domain of Y_2_O_3_ and MgO was distributed less uniformly and the average grain size of MgO was ~4 µm. Figure 8b shows that samples sintered with 160C-5S presented large residual pores and that the distribution of Y_2_O_3_ and MgO was not uniform either. The powder morphology of 160C-5S shows that the particle size is small, which may be related to the low hydrothermal temperature in Figure 8b. When calcining the powder prepared from the solution, the water released from calcination will adhere to the surface of the precursor, and the formation of oxygen bridge bonds will lead to agglomeration. Under 160 °C, the hydrothermal temperature significantly affects the crystallinity of the product, and MgO does not play a role anymore in preventing the growth of Y_2_O_3_ grains. However, when the nanocomposite is sintered with 190C-5S powder, as shown in Figure 8c, fine Y_2_O_3_ grains with a size of ~1 µm and MgO grains with a size of ~620 nm can be achieved, and the two phases present homogeneous distributions without a third phase or micropores being found. Figure 8d shows that for the 220C-5S powders, the MgO and Y_2_O_3_ in the ceramic were not uniformly dispersed and visible pores could be detected.

According to Figure 9a–e, microstructures of the Y_2_O_3_-MgO composite ceramics were observed. As shown in Figure 9a, MgO and Y_2_O_3_ phases had a uniform phase distribution. The dark particle of Point 1 was Y_2_O_3_, and the bright particle of Point 2 was MgO phase. Figure 9b shows selected area electron diffraction (SAED) patterns that indicate the crystallinity of the Y_2_O_3_. The represented diffraction rings are indicated in relation to the cubic Y_2_O_3_ in accordance with the pdf card (JCPDS No. 01-079-1717). The characteristic patterns of (022), (222), (211) and (200) of the Y_2_O_3_ could be clearly observed. Figure 9c shows high-resolution (HR) TEM images of Point 1. Figure 9d shows SAED patterns indicating the crystallinity of the MgO. The represented diffractions are indicated in relation to the cubic MgO in accordance with the pdf card (JCPDS No. 01-089-7746), and the characteristic patterns of (111), (022) and (200) of the MgO could be clearly observed. Figure 9e shows high-resolution (HR) TEM images of Point 2. Based on the TEM observation, MgO-Y_2_O_3_ composite ceramics are well-crystallized.

## 4. Conclusions

MgO-Y_2_O_3_ composite nanopowders with a uniform element distribution and particle size of about 125 nm were synthesized by hydrothermal method combined with the addition of MgO and Y_2_O_3_ crystal seeds. By comparing the specific surface areas and SEM morphologies with the seed contents of 0, 1, 5 and 10 hydrothermal treatments at 230 °C, we found that the powders with a seed content of 5% presented a better uniformity and dispersion. The hydrothermal temperature influencing the morphologies and particle size of powders was further studied, and the results show that under 190 °C, nanopowders with less agglomeration could be obtained. 160C-5S, 190C-5S, 220C-5S and 230C-5S MgO-Y_2_O_3_ ceramics were then fabricated by HP technique, and the 190C-5S (adding 5% seeds under 190 °C hydrothermal treatment) displayed a homogenous two-phase microstructure, smaller grain size and fewer pores. In the future, it is necessary to explore the feasibility of relevant scale-up experiments and the stability of the prepared nanopowders and ceramics, particularly when used at high temperatures.

## Figures and Tables

**Figure 1 materials-16-00126-f001:**
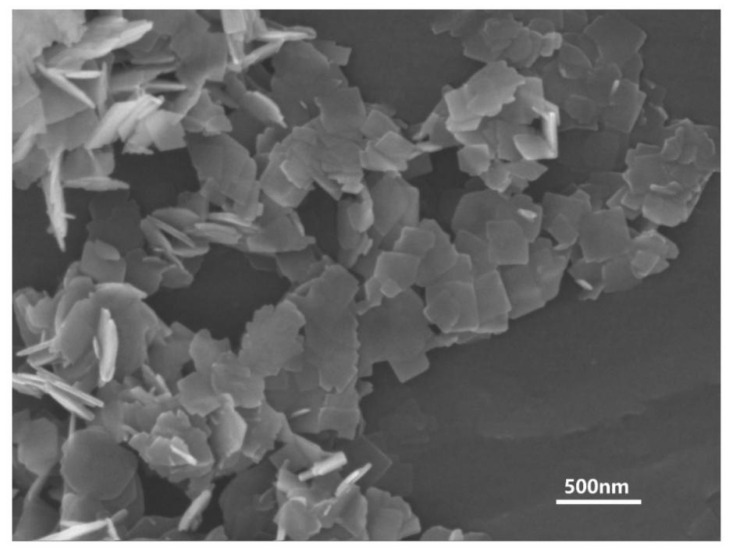
The SEM photograph of the hydrothermal precursors.

**Figure 2 materials-16-00126-f002:**
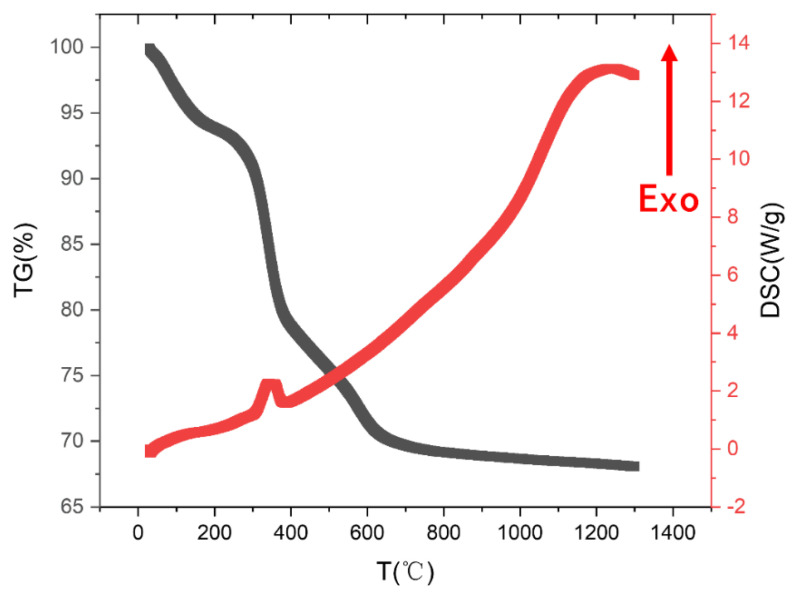
Plot of TG-DSC data for the precursors of hydrothermal powders. The initial amount of the sample was 8.767 mg.

**Figure 3 materials-16-00126-f003:**
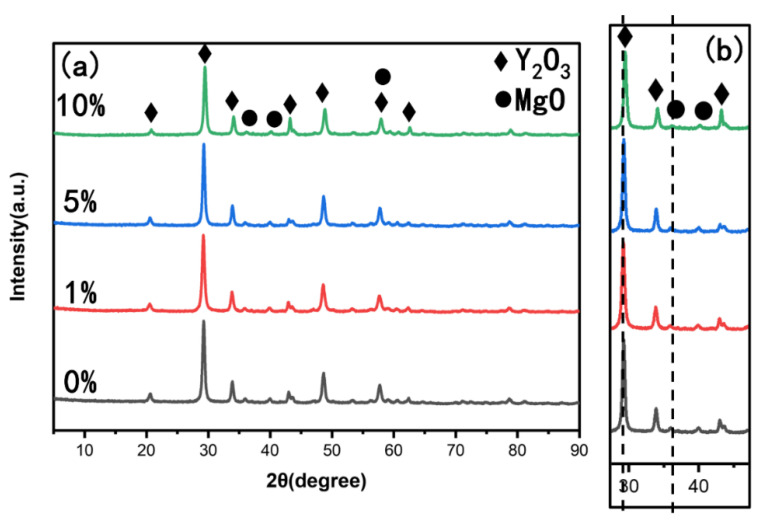
(**a**) X-ray diffraction patterns of powders derived from 230 °C for 2 h with different mass percentages of seed of 0%, 1%, 5% and 10%; (**b**) the details of the XRD patterns between 27° and 47°.

**Figure 4 materials-16-00126-f004:**
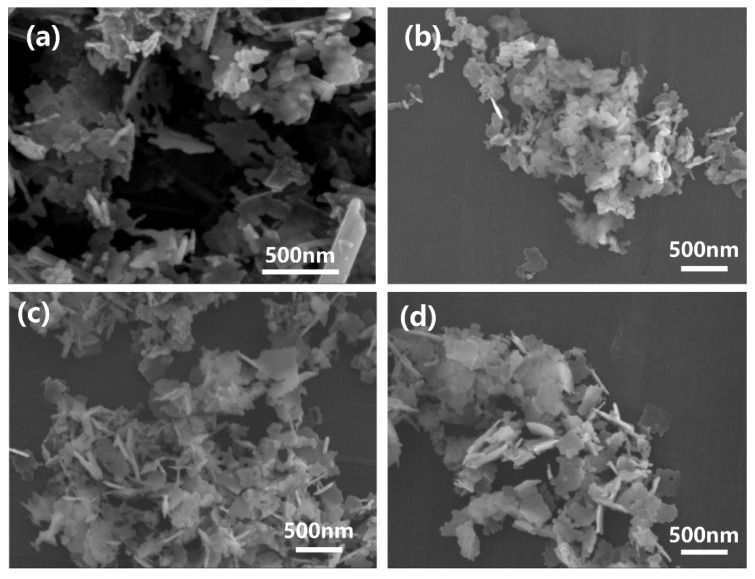
SEM images of MgO-Y_2_O_3_ nanopowders, with different seed contents of (**a**) 0%, (**b**) 1%, (**c**) 5% and (**d**) 10%, synthesized at a hydrothermal heat treatment temperature of 230 °C.

**Figure 5 materials-16-00126-f005:**
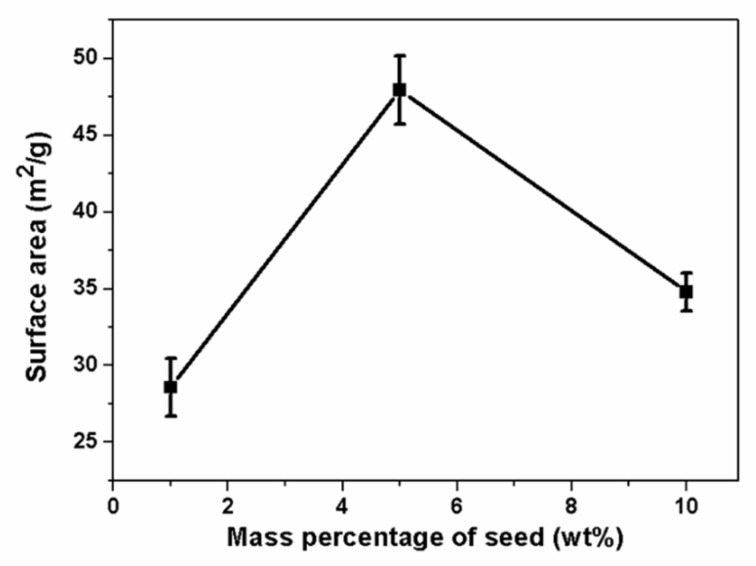
Specific surface area of the hydrothermal powders with different mass percentages of seeds treated at 230 °C.

**Figure 6 materials-16-00126-f006:**
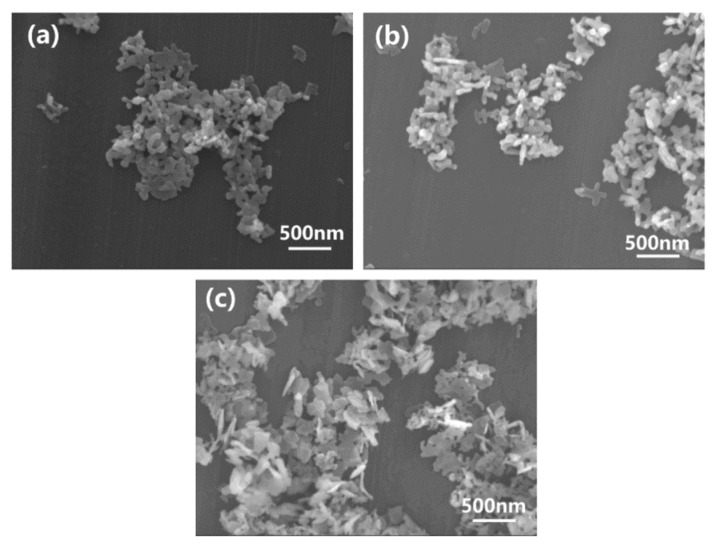
SEM images of MgO-Y_2_O_3_ nanopowders synthesized with 5% seeds with different heat treatment temperatures of (**a**) 160 °C, (**b**) 190 °C and (**c**) 220 °C (For comparison, see Figure 4c for 230 °C).

**Figure 7 materials-16-00126-f007:**
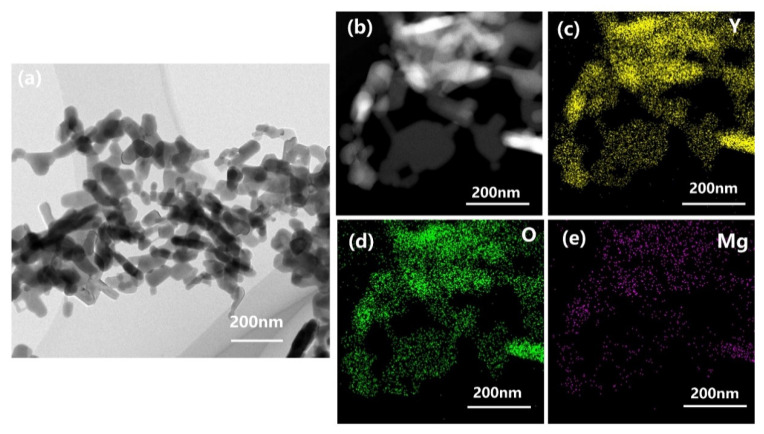
(**a**) TEM micrograph of hydrothermal MgO-Y_2_O_3_ powders at 190 °C with 5% seeds; (**b**) bright-field TEM image and EDS micrographs of (**c**) Y, (**d**) O and (**e**) Mg of hydrothermal MgO-Y_2_O_3_ powders at 190 °C with 5% seeds.

**Figure 8 materials-16-00126-f008:**
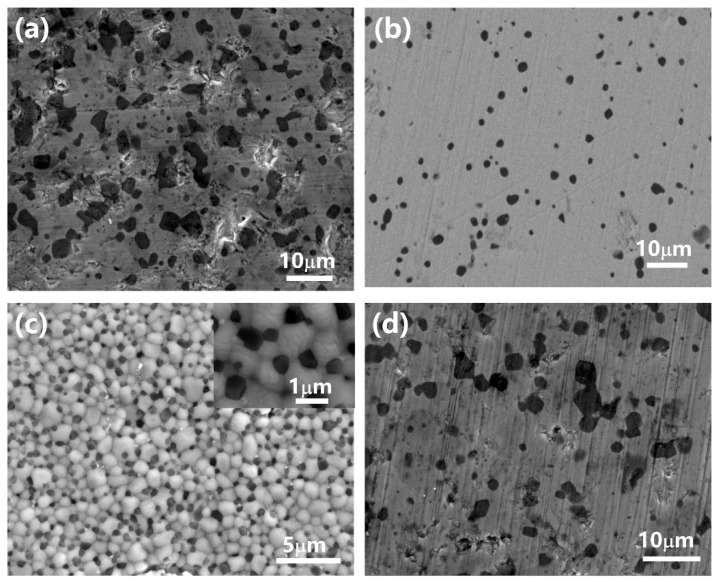
SEM micrographs of the MgO-Y_2_O_3_ sintered by HP method using the hydrothermal powders of (**a**) 230C-0S, (**b**) 160C-5S, (**c**) 190C-5S and (**d**) 220C-5S, respectively.

**Figure 9 materials-16-00126-f009:**
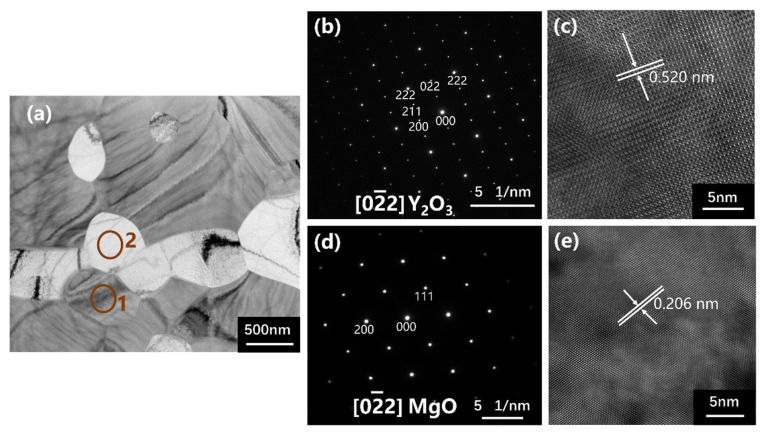
(**a**) The TEM micrograph of fracture surfaces of MgO-Y_2_O_3_ ceramic; (**b**,**c**) the SAED pattern and TEM image corresponding to Area 1 marked in (**a**); (**d**,**e**) The SAED pattern and TEM image corresponding to Area 2 marked in (**a**).

## Data Availability

Not applicable.

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
