# Peer review of "Fabrication of MgO-Y2O3 Composite Nanopowders by Combining Hydrothermal and Seeding Methods"

_materials, 2022, doi:10.3390/ma16010126_

Round 1
Reviewer 1 Report
The present manuscript Title: “Fabrication of MgO-Y2O3 Composite Nanopowders by Combining Hydrothermal and Seeding Methods" deals with the preparation and characterizarion of MgO-Y2O3 nanopowders. The introduction and background are given the premise of the manuscript. The results are consistent with the data and figures presented in the manuscript. While I believe this topic is of great interest, I think it needs minor revision before it is ready for publication. So, I recommend this manuscript for publication with minor revisions.
1) It would be better if the purpose of synthesizing the MgO-Y2O3 composite material and the places where it is used are mentioned in a little more detail.
2) The novelty of the research should be emphasized.

Reviewer 2 Report
The authors reported the fabrication of MgO-Y2O3 composite nanopowders by hydrothermal/seeding methods toward transparent ceramics. Different MgO-Y2O3 seeds contents have been discussed on the morphology of prepared nanoparticles and HIPed composite ceramics. The presented contents may give new ideas about making fine particles for transparent ceramics, I recommend it to be published on Materials after major revision, some questions and comments are listed below:
1. The main argue is about the less production amounts of powders by hydrothermal method and the small size of prepared transparent ceramics. Could the authors make some comments on this issure?
2. Fig. 8, judging from the SEM micrographs of HIPed sample, there are many micro-pores existed in the microstructure, which should be the main scattering centers for transparent ceramics? Could the author show the picture and in-line optical transmittance of the prepared ceramics? That maybe the main target for future applications, as mentioned in the introduction part.
Reviewer 3 Report
Please read and “fully” address the comments listed below:
1. The ABSTRACT is not written in a logical order. Start with an overview of the topic and a rationale for your paper. Describe the methodology you used and the general outline of the manuscript. Also, in the end, state the result in more detail (i.e., provide some numbers).
2. The novelty of your work is still unclear to the reader, which should be further detailed both in the Abstract and Introduction.
3. For XRD analysis of powder specify step size, dwell time, current, and voltage of analysis.
4. Apart from XRD analysis, please perform the XRF analysis (to find elemental composition of the powder).
5. Add error bar to Fig. 5 and provide more explanation on how the specific surface area of the hydrothermal powder changes with different mass percentages of seed.
6. Please fully explain the process used for mixing and casting the composite specimens.
7. Please provide more explanation for this sentence: Page 4, Line 138: “However, our careful analysis showed that a difference in width of the X-ray peaks of the synthesized particles can be detected.”
8. Similarly, provide more explanation for this sentence: Page 8, Line 236: " The powder morphology of 160C-5S shows that the particle size is small, which may be related to the low hydrothermal temperature of Fig 8b.”
9. It was mentioned that the Y(NO3)3·6H2O and Mg(NO3)2·6H2O with Y2O3 to MgO volume ratio (Y2O3:MgO = 50:50) were dissolved in distilled deionized water. Basically, (in general) the distilled water can react with MgO (periclase) forming Mg(OH)2, i.e., called Brucite, which can potentially results in volume expansions or “Unsoundness”. Therefore, please create a small section in the manuscript explaining that in future you have a plan to measure the volume stability of MgO-Y2O3 Composites. Also, please define the “Soundness” terminology (i.e., a term used to describe cement paste specimens that do not exhibit cracks, disintegration, or other flaws, that result from excessive volume change) in your paper. Moreover, the following papers are excellent examples of measuring soundness of concrete (please reference them in your paper):
• Mehta, P. K. (1978). History and status of performance tests for evaluation of soundness of cements. In Cement Standards—Evolution and Trends. ASTM International.
• Kabir, H., Hooton, R. D., & Popoff, N. J. (2020). Evaluation of cement soundness using the ASTM C151 autoclave expansion test. Cement and Concrete Research, 136, 106159.
10. Conclusion: Can authors highlight future research directions and recommendations? Also, highlight the assumptions and limitations (e.g., 1-2 shortcoming(s) of the present study). Besides, recheck your manuscript and polish it for grammatical mistakes (you can use “Grammarly” or similar software to quickly edit your document).